# Application of Machine Learning for Data with an Atmospheric Corrosion Monitoring Sensor Based on Strain Measurements

**Taisei Okura [1], Naoya Kasai [1,*], Hirotsugu Minowa [2] and Shinji Okazaki [3]**

1 Graduate School of Environment and Information Sciences, Yokohama National University, Yokohama 240-8501, Japan; okura-taisei-dw@ynu.jp
2 The Faculty of Business Administration, the Department of Business Administration, Okayama Shoka University, Okayama 700-8601, Japan; minowa@po.osu.ac.jp
3 Graduate School of Engineering, Yokohama National University, Yokohama 240-8501, Japan; okazaki-shinji-yp@ynu.ac.jp
* Correspondence: kasai-naoya-pf@ynu.ac.jp; Tel.: +81-45-339-3979

**Abstract:** Machine learning methods were applied to data with an atmospheric corrosion monitoring sensor based on strain measurements to improve the evaluation accuracy of the thickness reduction of a low-carbon steel plate due to atmospheric corrosion. Monitoring data used in this study were taken in a previous study using active–dummy strain gauges for corrosion product experiments. Values measured by the gauges before inducing corrosion via saltwater treatment of the test piece and reference data of the thickness reduction in a reference test piece were used for training data. By using the trained machine learning methods, the errors for the outputs of the machine learning models were smaller than those for the evaluation in monitoring data of our previous study.

**Keywords:** ACM sensor; steel test piece; strain measurement; machine learning; supervised learning

## 1. Introduction

To prevent serious accidents from occurring due to the aging of buildings and infrastructure, and to maintain safety, corrosive factors in the environment must be accurately measured and the appropriate maintenance performed. Studies on corrosion have been conducted by various researchers. Perveen et al. [1] studied a printed circuit board with a wireless, inductively coupled corrosion potential sensor to monitor the corrosion of steel-reinforced concrete civil infrastructure. Almubaied et al. [2] investigated the evolution of corrosion detected using fiber Bragg grating (FBG) sensing techniques and found a correlation between the FBG wavelength shift and the corrosion percentage of the reinforcement material. Hassan et al. [3] studied an optical sensor for monitoring the corrosion of reinforcement rebar. This method offers a real-time and inexpensive technique for applications involving remote monitoring. Hu et al. [4] developed a corrosion sensor for steel based on an iron film-coated optical fiber polarizer which can be used for monitoring the early stage of steel corrosion by building the relationship between the corrosion status and the polarization characteristics. Chen et al. [5] investigated modifying the relationship between the Bragg wavelength shift and the axial strain of the FBG to make highly accurate predictions of the level of corrosion of a steel bar embedded in concrete. Al Handawi et al. [6] researched a strain-based FBG corrosion sensor and tested the sensor on mild steel specimens to demonstrate its capability to measure the corrosion rate with real-time monitoring. Shitanda et al. [7] developed an electrochemical sensor for monitoring the corrosion of a circuit board and demonstrated its usability for the detection of sulfur gas and high humidity.

Some researchers have worked on atmospheric corrosion monitoring for many years by using electrochemical methods such as electrochemical noise (EN) and electrochemical impedance spectroscopy (EIS). Xia, Dahai et al. [8] established a portable EN monitoring system, designed two electrochemical probes, and concluded that EN can be used as

a new method to identify the form of corrosion resistance in atmospheric conditions. Nishikata et al. [9] monitored weathering steel corrosion under natural atmosphere by an EIS. Xia, D. H et al. [10] reviewed electrochemical probes and sensors used to detect and monitor atmospheric corrosion and concluded that the gap between adjacent electrodes and electrode sides affect the electrochemical measurement results.

Additionally, some researchers have studied atmospheric corrosion monitoring (ACM) sensors. Shinohara et al. [11] developed an ACM sensor consisting of an Fe-Ag galvanic couple and evaluated the corrosiveness of outdoor and indoor environments. Mizuno et al. [12] investigated an ACM sensor for automotive parts that was based on an analysis of the correlation between the output of ACM sensors and the corrosion rates of actual materials.

Our previous studies have proposed an ACM sensor using strain measurements [13–16]. Nining et al. [15] studied an ACM sensor using strain measurements with the active–dummy method and reported that the measured signals of the test piece corresponded to its thickness reduction. However, these signals contained errors induced by temperature changes and corrosion products.

To improve the evaluation of the corrosion caused in an environment, some researchers have focused on applying the machine learning methods to the study of corrosion evaluation. Aghaaminiha et al. [17] modeled measurements of the corrosion rates of carbon steel as a function of time and found that the sensitivity of corrosion rates to changes in the environmental variables were well predicted by a trained random forest model. Pei et al. [18] studied predicting instantaneous atmospheric corrosion using an Fe/Cu-type galvanic corrosion sensor and a random-forest-based machine learning approach.

As mentioned above, by using machine learning, the data concerning corrosion could be used to make more accurate corrosion evaluations. This study applied machine learning methods to the data obtained from our previous study and evaluated the amount of thickness reduction of a low-carbon steel test piece.

## 2. Methods

### 2.1. ACM Sensor Based on Strain Measurements

When the test piece is damaged by corrosion, the thickness of the test piece decreases and the strain at the surface changes. The strain gauge attached to the test piece measures the strain and calculates the thickness reduction based on the following theory.

Figure 1 shows a test piece with thickness $h$, bending moment $M$, radius of curvature $\rho$, and corrosion-induced thickness reduction $\Delta h$ under the bending moment. The A-A plane is subjected to compressive strain and the B-B plane is subjected to tensile strain. The length of the O-O plane after bending is equal to the length of the O-O plane before bending. The O-O plane after bending is thus $\rho d\theta$, as shown in Figure 1a. Since the length of the A-A plane after bending is $\left(\rho - \frac{h}{2}\right)d\theta$, the strain $\varepsilon$ induced in the A-A surface can be expressed by Equation (1).

$$\varepsilon = \frac{\left(\rho - \frac{h}{2}\right)d\theta - \rho d\theta}{\rho d\theta} = -\frac{h}{2\rho} \tag{1}$$

$\Delta h$ is generated due to corrosion, as shown in Figure 1b, and the B-B and O-O planes move to the B′-B′ and O′-O′ planes. Since $\Delta h$ is negligibly small compared to $\rho$, the radius of curvature $\rho - \Delta h/2$ can be approximated by $\rho$. The change in strain is expressed by Equation (2).

$$\Delta\varepsilon = -\Delta h/2\rho \tag{2}$$

Corrosion-induced thickness reduction $\Delta h$ can be obtained by rewriting Equation (2) to Equation (3).

$$\Delta h = -2\rho\Delta\varepsilon \tag{3}$$

The decrease in the test piece thickness can be measured by the change in strain at the constant radius of curvature $\rho$. Therefore, based on Equation (3), we can monitor the level of corrosion by measuring the strain with a sensor.

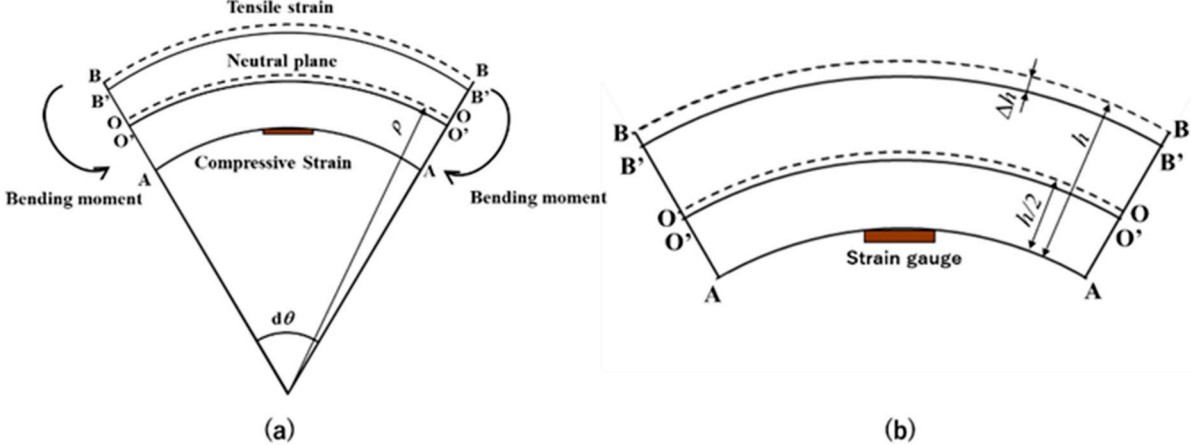

**Figure 1.** Schematic diagram of the test piece indicating (**a**) the dimensions and curvature of test specimen, and (**b**) the decreases in thickness and the position of strain gauge.

*2.2. Data Collection with an ACM Sensor*

The test piece was 95 mm in length, 45 mm in width, and 0.5 mm in thickness. The material was low-carbon steel and the total area exposed to corrosion was 1350 mm$^2$. A schematic diagram of the test piece is shown in Figure 2. The ACM device also had a base and cover with $\rho$ = 430 mm, as in reference [15]. The test piece was placed in the apparatus. Active and dummy strain gauges were installed on the back surface, as shown in Figure 2. A full bridge with 2 active and 2 dummy gauges was employed to enhance the accuracy of the measurement. In addition, to cancel noise from the strain measurement circuit, two identical strain measurement circuits were manufactured, and one of them was used as a dummy circuit [14,15]. The active gauges of the active circuit detected strain based on changes in the thickness of the test piece, temperature changes, and the formation of corrosion products. The dummy gauges of the active circuit detected the strain changes only due to the temperature. By taking the difference between the active and dummy gauge values, the change in strain due to the temperature can be removed and these differential values were taken to be the actual amount of thickness reduction due to corrosion in the test piece.

Corrosion monitoring data were collected for 83 days and the measurements are plotted in Figure 3. The time interval was 10 min and there were 11,962 data points in total. In Figure 3, the left vertical axis is the strain measured by the sensor and the right vertical axis is the temperature. The red circles are the decreases in the thickness of the reference coupon test piece. Since the reference test piece was installed near the sensor, it was assumed that the conditions were the same as that of the test piece installed near the sensor. In addition, the reference test piece underwent the same saltwater treatment as the test piece of the ACM sensor. The blue line is the active gauge data, the green line is the dummy gauge data, and the yellow line is the temperature. The active gauge strain decreased until about the 40-day mark, after which it then showed an increasing trend. On the other hand, the dummy gauge strain was gradually decreasing overall. The temperature was gradually increasing due to seasonal factors with the daily range. The black line is the difference between the active and dummy values, and these data were used as the monitoring data. In our previous study [14], three stages of the black line were described. The first stage was considered the initial condition and preceded the application of a saltwater spray. This stage lasts for 15 days and during this time, the reduction in thickness is almost always measured to be 0 mm, as in Figure 3. The second stage defines the conditions from day 15 to 40 after spraying the salt water, in which corrosion products

are generated, as shown by the negative trend in the measured strain. During this time, the test piece thickness is increased by the accumulation of corrosion products. The third stage demonstrates corrosion progression. It showed a positive trend in the measured strain, which indicates that the test piece thickness decreased due to corrosion. This type of data therefore poses two problems for accurate corrosion monitoring. The first is the negative trend of the second stage. According to Equation (3), strain increases when thickness decreases; however, the strain decreased. The second is the noise in the overall data. The goal of our research is to eliminate these problems and produce more accurate predictions of the actual corrosion rate by applying machine learning methods.

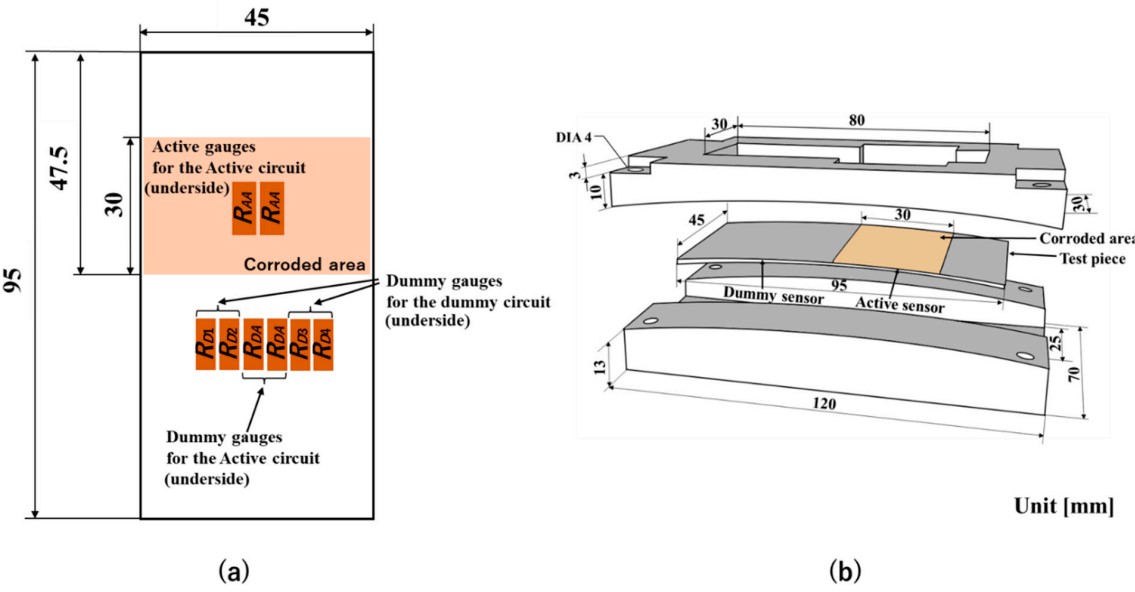

**Figure 2.** Schematic diagram of the ACM device and test piece indicating (**a**) the configuration of strain gauges attached to the piece and (**b**) the apparatus for atmospheric corrosion sensor using strain measurements [14].

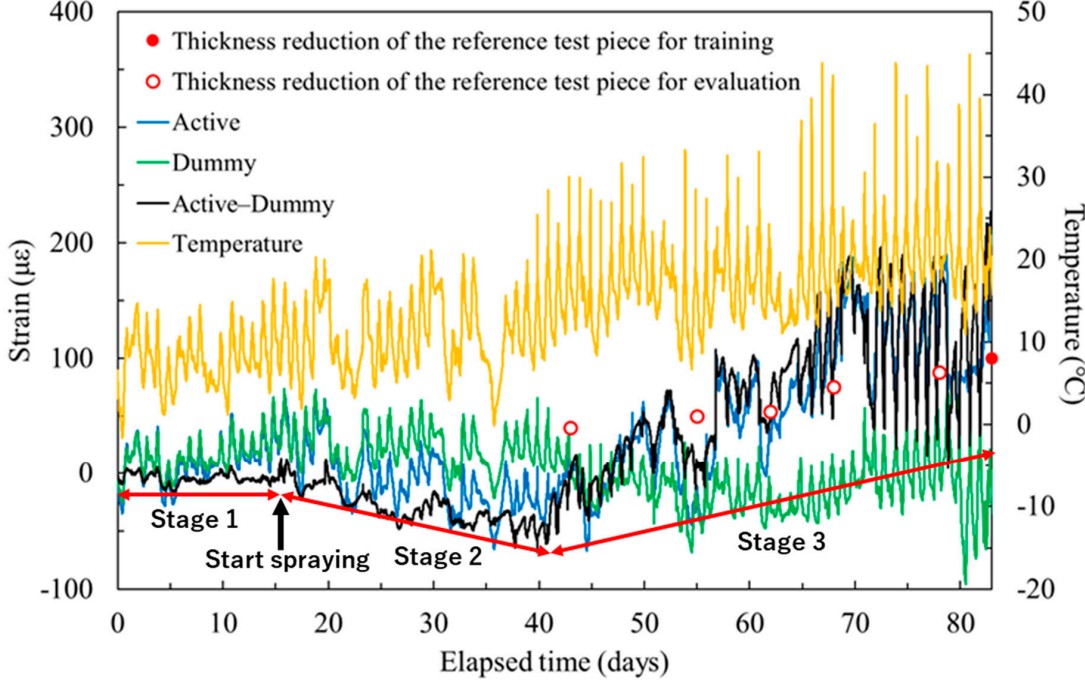

**Figure 3.** Corrosion monitoring data collected over 83 days [14].

### 3. Machine Learning Approach to Predicting Corrosion

*3.1. Supervised Learning*

Supervised learning was used in the present study. A portion of the measurement data from our previous study was used as training data for the model. Initially, the measurements made during the first stage, in which it was clear that the strain was zero, were used as training data. The number of these training data points was 2161. However, from these data alone, we were unable to train the model to correctly evaluate the thickness reduction, which should have increased over time; therefore, the final thickness reduction was also entered as one training data point corresponding to the endpoint of the measurements. According to the known thickness reduction of the reference test piece, the final value of the strain was about 100 με. This value was chosen as a training data point for the corrosion progression region and the details of all the training data are summarized in Table 1.

**Table 1.** Training data characteristics.

| Description | Value | Number of Data Points |
|---|---|---|
| Strain before spraying | 0 | 2161 |
| Final strain | 100 | 1 |
| Total number | | 2162 |

In this study, a neural network and a support vector machine were used as the machine learning methods. The software Orange Canvas [19], which is a software that allows for visual programming by combining widgets, was used for the training and validation of the machine learning model.

*3.2. Data Preprocessing*

Before training the model, preprocessing of the monitoring data was performed. By applying a four-day moving average, noise due to abrupt temperature changes was removed, as shown in Figure 4. The data with this moving average applied were used as features in the machine learning training to create a model that can evaluate the amount of thickness reduction of the test piece after 83 days.

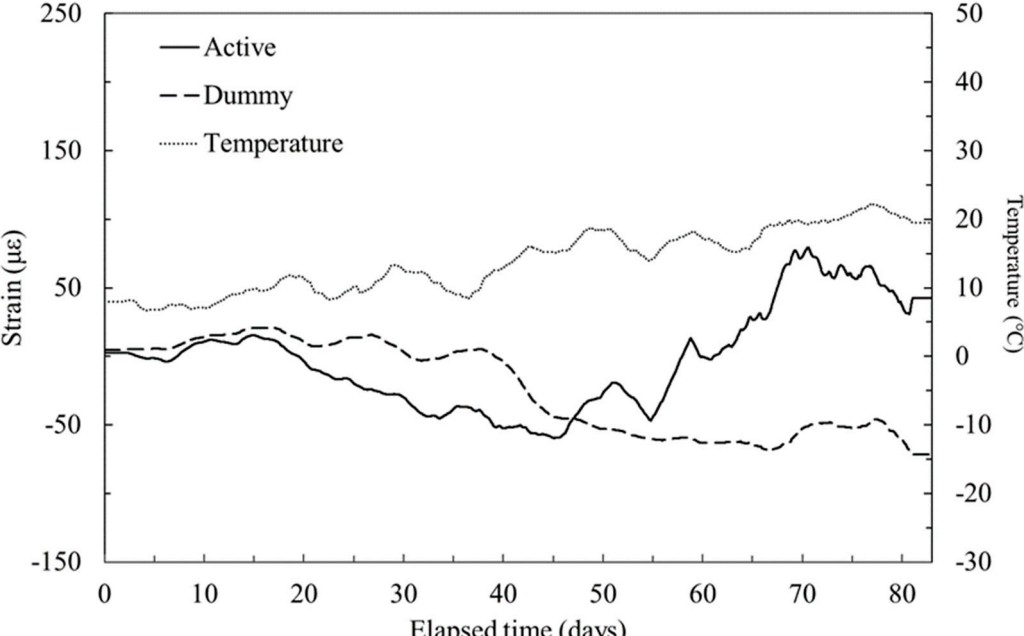

**Figure 4.** Four-day moving average of the active gauge, dummy gauge, and temperature data.

### 3.3. Model Validation

We validated the performance of the machine learning model. The data from the first stage were divided into two sets and used as training data and test data. The moving average data for the active gauge, the dummy gauge, and the temperature were used as the features. The value of the first stage was considered to be zero because the first stage occurs before spraying the saltwater, that is, the thickness reduction is zero. To compare the accuracy of the evaluation results, the root mean square error (*RMSE*) was used. The RMSE is given by the formula in Equation (4). Here, $f_k$ is the value predicted by machine learning, $y_k$ is the ideal value of the monitoring data, that is, $f_k = 0$ mm, and $n$ is the number of data points.

$$RMSE = \sqrt{\frac{1}{n} \sum_{k=1}^{n} (f_k - y_k)^2} \tag{4}$$

The machine learning models predicted values of mostly zero and a small error was obtained. Root mean square errors between the data source and reference data are shown in Table 2. We then proceeded to apply the trained models to predict the thickness reduction over a period corresponding to the entire monitoring duration.

**Table 2.** Root mean square error of the data before corrosion generates.

| Data Source | Root Mean Square Error $[\mu m]$ |
| --- | --- |
| Neural network prediction | 0.5 |
| Support vector machine prediction | 0.0 |
| Monitoring data | 1.4 |

### 3.4. Evaluation Result

The level of corrosion over the entire monitoring period was evaluated using two machine learning models, namely a neural network and a support vector machine trained with the data described in Section 3.1. The neural network employed in the study had 20 layers. The activation function was ReLu and the solver was Adam. The architecture of the neural network was set up after examining several patterns in the preliminary analysis. In the support vector machine, the regression type was used. The linear function was set as the kernel function. The cost was 1 and the regression loss was 0.1. The data for the four-day moving averages of the active gauge strain, dummy gauge strain, and temperature, as well as the elapsed time since the start of the experiment, were used as features. The RMSEs between the reference test piece data and the evaluation results of the machine learning models are summarized in Table 3, and the relationships between the reference piece data and the results predicted by both the neural network and the support vector machine are shown in Figure 5. The black circles in Figure 5 indicate the amount of thickness reduction of the reference test piece, the solid line is the prediction result of the neural network, and the dotted line is the prediction result of the support vector machine. The values during the first 15 days were predicted to be 0 μm by the neural network and had a small error based on the thickness reduction values of the reference test piece. After 15 days, there were temporary decreases, but the overall trend was a continued increase and the error also became smaller. After 40 days, there were again temporary decreases, but the overall trend was also a continued increase. On the other hand, the values for the first 15 days predicted by the support vector machine were not 0 μm. The support vector machine predictions showed an increasing trend until day 20 and then decreased until day 40.

In the present study, we were able to use a neural network to obtain predicted results that were closer to the amount of thickness reduction in the reference test piece than the data obtained by ACM sensors. These results suggest that neural networks can be suitable for improving the accuracy of predicted atmospheric corrosion.

**Table 3.** Root mean square error of the thickness reduction between reference values, predicted values, and monitoring data.

| Data Source | Root Mean Square Error [μm] |
| --- | --- |
| Neural network prediction | 8.8 |
| Support vector machine prediction | 15.7 |
| Monitoring data | 31.1 |

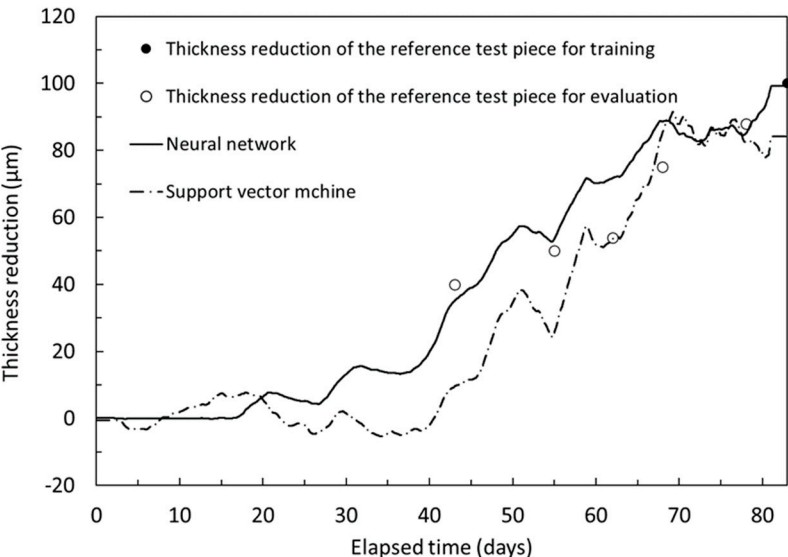

**Figure 5.** Prediction results for the neural network and support vector machine models. Black circles indicate the amount of thickness reduction of the reference test piece.

### 4. Conclusions

In this study, we applied machine learning to an ACM sensor using strain measurements with an active–dummy method. We used a neural network model and a support vector machine model as regression models to predict the total thickness reduction due to corrosion. The model was trained using the following features: the active gauge strain, the dummy gauge strain, temperature, and elapsed time. The active gauge, dummy gauge, and temperature data had four-day moving averages applied to remove noise due to abrupt temperature change. The performance of the machine learning models was evaluated with validation data corresponding to the 15 days before the corrosion started. It was found that the errors between the predicted values by the machine learning models and the test data were small, and the models were deemed adequate. The models were then used to predict the thickness reduction over the entire monitoring period and these values were compared to those of the reference test piece. The errors for the outputs of the machine learning models were smaller than those for the evaluation in monitoring data of our previous study [15]. The results of this study demonstrate that an ACM sensor with applied machine learning may have the potential to be applied in the maintenance of, for example, steel structures or buildings.

**Author Contributions:** Conceptualization, T.O., N.K. and H.M.; Methodology, T.O., N.K. and H.M.; Software, T.O. and H.M.; Validation, T.O., N.K. and S.O; Formal analysis, T.O.; Investigation, T.O. and N.K.; Resources, N.K.; Data curation, T.O., N.K. and H.M.; Writing—original draft preparation, T.O.; Writing—review and editing, T.O. and N.K.; Visualization, T.O.; Supervision, N.K., H.M. and S.O.; Funding acquisition, N.K. All authors have read and agreed to the published version of the manuscript.

**Funding:** This research received no external funding.

**Institutional Review Board Statement:** Not applicable.

**Informed Consent Statement:** Not applicable.

**Data Availability Statement:** The data that support the findings of this study are available from the corresponding author, Kasai, N. upon reasonable request.

**Conflicts of Interest:** The authors declare no conflict of interest.

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
