# Peer review of "Application of Machine Learning for Data with an Atmospheric Corrosion Monitoring Sensor Based on Strain Measurements"

_metals, doi:10.3390/met12071179_

Round 1

Reviewer 1 Report

This paper proposed using two machine learning methods are used to study the output error of
corrosion product experiments active-dummy strain gauges, which is smaller than the error of
monitoring data evaluation in the previous study, and the authors give a detailed description of the
experimental steps and give a detailed diagram and the data results of the final training using
machine learning, this is interesting. Overall the paper is fairly well written. The following revisions
are suggested:
1. this paper uses two machine learning algorithms, and their application scope should be different.
I suggest that the authors should further explain why we choose these two algorithms and give
some description of them.
2. In the third chapter, the first section describes the use of the training data, described as training
data points to 2161, and the endpoint is also as the training data points, but the second chapter
describes the amount of data for 11962, 2162 training data points of the definition of the registry
is a value of 0 points, the author should be clear whether the input training data points is the only
value of 0 points, And whether the addition of the endpoint makes a difference in the case of a
large number of inputs.
3. In this article, RMSE is used in two places, one of which is the root mean square error of the
thickness reduction and the other is the root mean square error of the thickness reduction between
reference values, and predicted values, I suggest further clarification of whether these two measures
are used to co-confirm the results or to compare them in two different ways.
4. There are minor grammatical mistakes and typos should be corrected.

Reviewer 2 Report

There are the following comments and questions:

      You specify the features values explicitly (active gauge strain, dummy gauge strain, temperature, and elapsed time (lines 217-218), but do not specify the target parameter. Is it thickness reduction or the strain?

2.      It is not entirely clear how the training set was constructed. It consisted (in accordance with Table 1) of 2,161 points with the strain was zero  and 1 point with the strain was 100 με.  If this is the case, then the training set is very unbalanced.

3.      What type of support vector machine was used: regression or classification? What are the hyperparameters of the support vector machine? How were they chosen?

4.      What is the architecture of the neural network? How did you set it up?

Reviewer 3 Report

This is a very interesting paper, as the author developed Machine Learning for Data With Atmospheric Corrosion Monitoring Sensor Based on Strain measurements. Overall the methods reported in this paper is novel and difinitely deserve to be published. The paper is well written, and is in good structure.

I suggest a minor revision.

(1) The reviewer has worked on atmospheric corrosion monitoring for many years by using electrochemical methods especially by using electrochemical noise (EN) methods. Therefore, ACM methods should be compared with other electrochemical methods such as EIS and EN. A recent critical review entitled "Electrochemical probes and sensors designed for time-dependent atmospheric corrosion monitoring Fundamentals, progress, and challenges" can be mensioned.

In addition, Digby Macdonald et al. has published a critical review in the Journal of Materials Science & Technology  (2022), this papar should also be mensioned.

2. what is the limitations of this methods? It should be discussed as well.
